# Application of the Discrete Element Method to Study the Effects of Stream Characteristics on Screening Performance

**Ali Davoodi** ***, Gauti Asbjörnsson, Erik Hulthén and Magnus Evertsson**

Department of Industrial and Materials Science, Chalmers University of Technology, 412 96 Gothenburg, Sweden; gauti@chalmers.se (G.A.); erik.hulthen@chalmers.se (E.H.); magnus.evertsson@chalmers.se (M.E.)
* Correspondence: ali.davoodi@chalmers.se

**Abstract:** Screening is a key operation in a crushing plant that ensures adequate product quality of aggregates in mineral processing. The screening process can be divided into the two sub-processes of stratification and passage. The stratification process is affected by the relative difference between various properties, such as particle shape, size distribution, and material density. The discrete element method (DEM) is a suitable method for analyzing the interactions between individual particles and between particles and a screen deck in a controlled environment. The main benefit of using the DEM for simulating the screening process is that this method enables the tracking of individual particles in the material flow, and all of the collisions between particles and between particles and boundaries. This paper presents how different particle densities and flowrates affect material stratification and, in turn, the screening performance. The results of this study show that higher density particles have a higher probability of passage because of their higher stratification rate, which increases the probability that a particle will contact the screen deck during the process.

**Keywords:** screen; DEM; discrete element method; classification efficiency; separation

## 1. Introduction

Screening is a key unit operation in a crushing plant that ensures adequate product quality of aggregates in mineral processing. The main purpose of screening is to classify particles based on their shape and size. Efficient screening can reduce operational costs by improving product structure, product quality, and energy efficiency [1].

During the screening process, many operational parameters can affect the performance of screening. These parameters can be divided into either machine parameters or stream characteristics. Machine parameters, such as the screen dimensions, deck material, vibration frequency, vibration amplitude, and inclination, are dependent on the installed unit and selected operational strategy. The stream characteristics include material properties such as the size distribution, shape, density, and flow rate of the material [2].

During operation, a material bed builds upon each deck. Particles that are smaller than the aperture and are in contact with the deck have a certain probability of passing through the aperture. Soldinger [3] proposed a mechanistic modeling approach to screening in which the granular flow was discretized along the deck and in horizontal layers to approximate horizontal and vertical transport. In the stratification process, the finer fractions change vertical placement with the larger fractions that are located in lower layers over a predefined time interval [4]. The passage rate is a function of the particle size distribution in the bottom layer of the bed, i.e., the particles that are in contact with the screen deck [4]. However, the model is not capable of approximating the change in the stratification rate as a function of the particle shape or density.



Granular materials with different properties exhibit different segregation behaviors, whether in a vibrating or flowing bed. The three most important material properties that can affect segregation are particle shape, size, and density [5].

Different models have been proposed for predicting segregation in granular flows. In a research study by Khakhar et al. [6], a model for the segregation flux in flowing layers was presented and validated by particle dynamics and Monte Carlo simulations for steady flow down an inclined plane. Bridgwater et al. [7] developed a theory and used a mathematical model to describe the segregation of granular material; the authors also developed a model with a segregation velocity based on separation due to particle size differences, and in the model proposed by Wang et al. [8], the particle movement was considered. In work by Clément et al. [9], experiments on the motion of single particles of different sizes in a bed were done by using an image-processing device. In a study by Xiao et al. [10], both the discrete element method (DEM) and experiments were used to model density segregation in a flowing material. Only a few works have been completed by using continuum methodologies. In work by Carvalho and Tavares [11], an efficient method was presented for solving the two-way coupling of DEM and a mechanistic breakage model, to describe continuous grinding in a pilot-scale mil.

The granular flow of particles in the screening process is dependent on the interactions between individual particles, and between particles and the screen deck, which can be difficult to quantify experimentally. Simulation has become a common tool in the design and optimization of the processing of granular materials [12]. Of the simulation methods that have been used recently, one is the discrete element method (DEM). DEM is a simulation method that is discontinuous, which is required in granular material simulations because of the highly discontinuous nature of these materials [13]. Several studies have been conducted using DEM simulations to analyze separation performance by studying different parameters and particle behavior during screening operation. For example, the effect of particle size distribution was studied by Jahani et al. [14], and the effects of the aperture shapes and materials of screen decks on the screening efficiency were studied by Davoodi et al. [15]. Additionally, the effect of different levels of acceleration was studied by et al. [16].

The objective of this paper is to study how different stream characteristics influence screening performance, in particular, the effect of the material density on stratification. In this paper, a model for the effect of the density on the stratification rate and performance is presented, and the approach is validated by comparing calibrated DEM simulations with an empirical model for vibratory screen. Table 1 shows the description of the parameters which have been used in this paper.

**Table 1.** Nomenclature.

| | | | |
|---|---|---|---|
| $F_n$ | Normal force, Equation (1), (N) | $M_{i,j,n_l}$ | The mass flow along the screen, Equation (4), (kg) |
| $K_n$ | Stiffness of the spring, Equation (1), (-) | $M_{down,i,j,n_l-1}$ | The downward flow, Equation (4), (kg) |
| $C_n$ | Viscoelastic damping constant, Equation (1), (m/s) | $M_{up,i,j,n_l}$ | The upward flow, Equation (4), (kg) |
| $V_n$ | The relative velocities, Equation (1), (m/s) | $M_{BP,i,j}$ | Mass flow of the material in the contact layer, Equation (4), (kg) |
| $F_t$ | Tangential force, Equation (2), (N) | $k_j$ | Passage rate parameter, Equation (4), (-) |
| $K_t$ | Stiffness of the spring, Equation (2), (-) | $\Delta t$ | Time step, Equation (4), (s) |
| $C_t$ | Viscoelastic damping constant, Equation (2), (-) | $\alpha$ | Slope of the screen deck, Equation (6), (Angle) |
| $V_t$ | The relative velocities, Equation (2), (m/s) | $v$ | Transport velocity of the particle along the screen deck, Equation (6), (m/s) |
| $f$ | Frequency, Equations (3) and (6), (Hz) | $R$ | Function of stroke, Equation (6), (mm) |

## 2. Methodology

### 2.1. DEM

#### 2.1.1. DEM Contact Model Theory

The DEM method uses a clear numerical structure to trace the motion of individual particles according to their interactions with each other and the system geometry (such as the screen deck and feeder), where a contact law is used to predict the velocities and rotations of the particles. There are various different contact models, such as the linear elastic and the Hertz–Mindlin contact models. However, the linear contact model may not be sufficiently precise for spherical particles. The Hertz–Mindlin contact model was developed to solve the contact behavior of spherical shapes. The Hertz–Mindlin contact model is a no-slip model with a linear spring-dashpot [17]. Figure 1 shows the interaction between two particles with frictional elements between the normal force and the tangential force [18]. The contact force between the impacting particles is split into a normal force $F_n$ and a tangential force $F_t$ as follows

$$F_n = -K_n \Delta x + C_n V_n \tag{1}$$

$$F_t = -K_t \Delta x + C_t V_t \tag{2}$$

where $\Delta x$ denotes the particle displacements in the normal and tangential directions and $V_n$ and $V_t$ denote the relative velocities. $K$ denotes the stiffness of the spring, and $C$ denotes the viscoelastic damping constant. If $F_t$ exceeds the limiting frictional force, then the particles will slide over each other, and the tangential force is calculated using the frictional coefficient $f$:

$$F_t = -f F_n \tag{3}$$

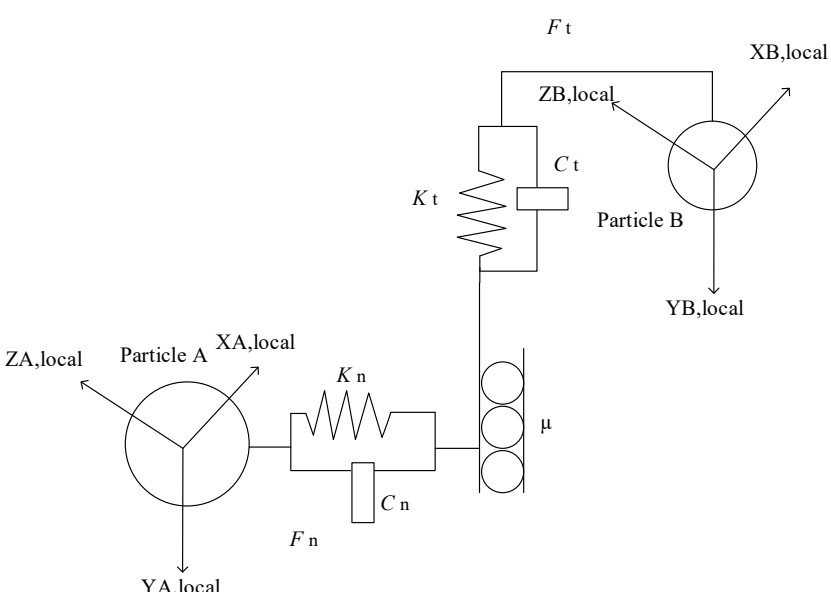

**Figure 1.** Graphic illustration of the Hertz–Mindlin contact model.

#### 2.1.2. Simulations

The test plan was designed to determine how different material densities and altering the feed rate affect the screening efficiency. A total of nine simulations were run: three simulations for each different material density. As Table 2 shows, the feed rate was the parameter that was varied in the simulation and had initial values of 4 kg/s, 5 kg/s, and 6 kg/s. The particle size distribution (PSD), which, when

generated for simulations are user-defined, meaning each of the particles are generated at different masses, as defined by the user. Figure 2 shows the PSD curve for the feed material in all simulations.

**Table 2.** of simulation parameters.

| Material Properties | Poisson's Ratio | Shear Modulus | Density |
|---|---|---|---|
| Particles (Low density) | 0.3 | 24 MPa | 2100 kg/m$^3$ |
| Particles (Medium density) | 0.3 | 24 MPa | 2500 kg/m$^3$ |
| Particles (High density) | 0.3 | 24 MPa | 2900 kg/m$^3$ |
| Screen (Steel) | 0.2 | 79 GPa | 7800 kg/m$^3$ |
| **Collision Properties** | **Coefficient of Restitution** | **Coefficient of Static Friction** | **Coefficient of Rolling Friction** |
| Particle-particle | 0.2 | 0.6 | 0.01 |
| Particle-screen (Steel) | 0.6 | 0.45 | 0.01 |
| **Machine Parameters** | | | |
| Screen aperture | 25 mm×25 mm and 10 mm×10 mm | | |
| Screen declination | 15° | | |
| Screen vibration | Sinusoidal translation, amplitude 6 mm, and frequency 13 Hz | | |
| Particle generation rate | For particles at 4 kg/s, 5 kg/s, and 6 kg/s | | |

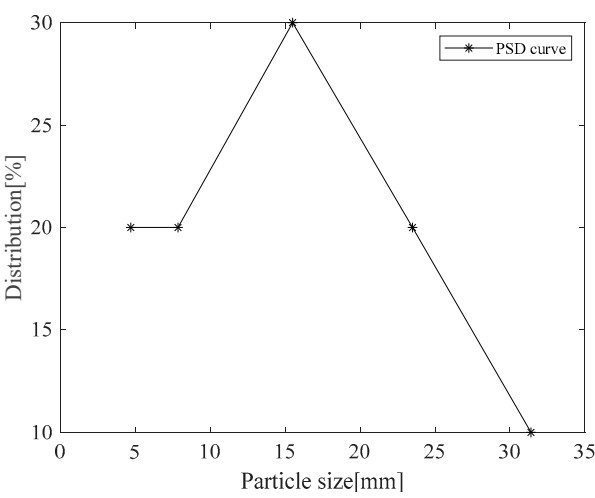

**Figure 2.** Particle Size Distribution (PSD) for feeding material.

The screen geometry that was used in simulations is based on a vibratory screen lab, a reduced-scale version of a real industry screen. The screen lab was used to minimize computing time by decreasing the geometrical area and the numbers of particles.

The next step was to set up the simulation. Particles with user-defined diameters were employed to match the planned simulations. All of the particles were non-spherical in shape, as very different results are obtained using one spherical particle in the simulations versus multi-sphere particles. Multi-sphere particles produce more realistic results because these particle shapes are closer to those of real particles [19]. Figure 3 shows the particle shape that was used in the simulations where all spheres had the same material properties. The material is a rock with three different solid densities of 2900 kg/m$^3$, 2500 kg/m$^3$, and 2100 kg/m$^3$. The separation of these particles may represent the essential sorting operation that occurs in a typical screening process. More generally, this model features the main characteristics of many industrial screening applications. Table 2 is a summary of the simulation parameters. The collision properties were calibrated by the method proposed by Quist [19].

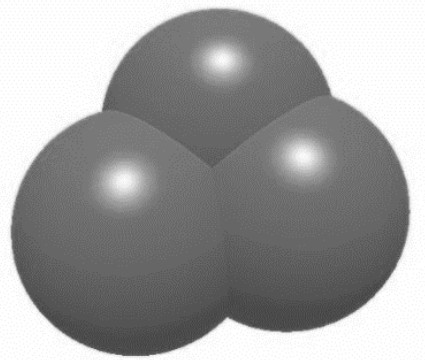

**Figure 3.** Three spheres particle used in simulations.

Each DEM simulation should achieve a steady-state before data extraction; that is, the total number of particles in the system should become stable after a given period. The average time for the overflow particles to travel along the entire deck, from the feed end to the discharge end, was 5 s in the simulations, and the simulations reached a steady-state after 6 s, as can be seen in Figure 4.

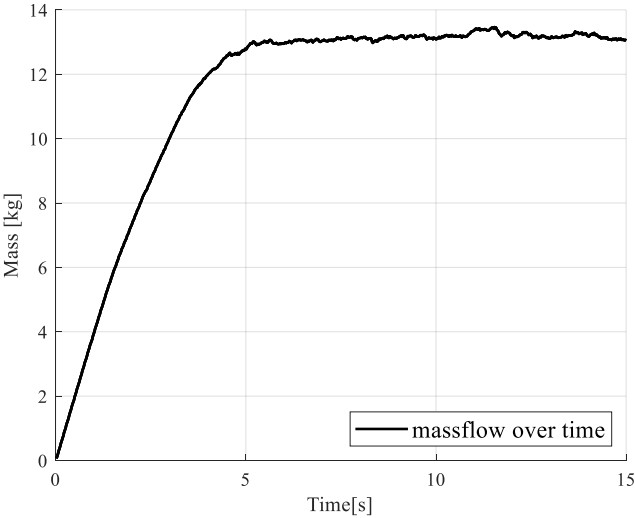

**Figure 4.** The mass flow over time in the simulation.

## 2.2. Laboratory Vibrating Screen

A laboratory-scale vibratory screen was constructed to control the screen motion, feed rate, and aperture size, and to enable the analysis of the particle size distribution. A schematic of the CAD geometry used in the simulations is shown in Figure 5. The screen was 1500 mm long and 300 mm wide.

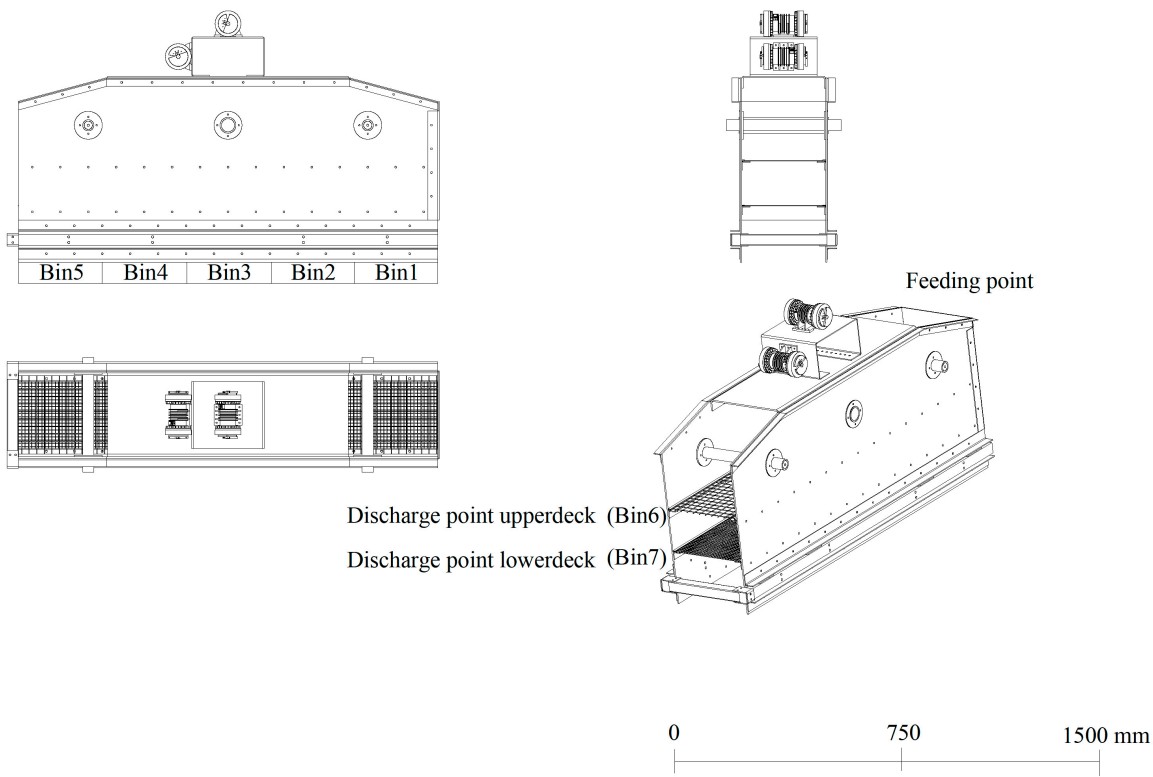

**Figure 5.** Schematic showing the CAD model used in the simulations.

## 2.3. Modeling

The screen model used in this study was based on the model proposed by Soldinger [20]. The material flow (*M*) was modeled as finite zones (*i*). Between the discrete zones, mass is transferred at each time step (Δ*t*) in different fractions (*j*) in a layer (*k*). The mass flow is governed by Equation (4):

$$M_{i+1,j,n_l} = M_{i,j,n_l} + M_{down,i,j,n_l-1} - M_{up,i,j,n_l} - M_{BP,i,j}k_j\Delta t \tag{4}$$

Every zone (*i*) was divided into two layers, an upper and a lower layer. Between these layers, the material was exchanged through stratification. Figure 6 shows an example of a layer model for stratification and passage process for one screen deck.

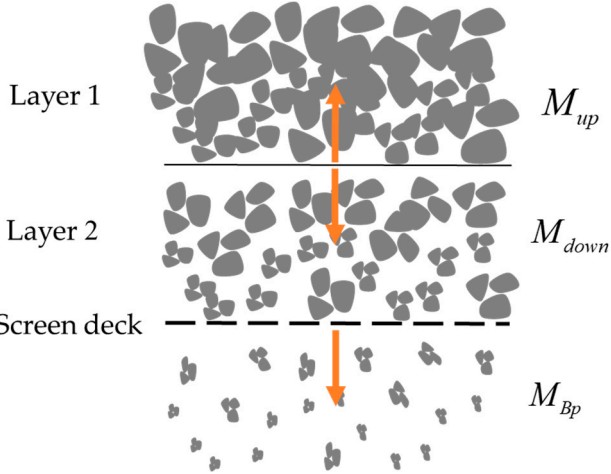

**Figure 6.** Layer model for stratification and passage process for one screen deck.

From the lower of these layers, material can pass through the screen apertures, which is determined by a passage probability ($k$), as described by Equation (5); which is determined by the average particle size in each fraction ($d_{50}$), the aperture size ($Ap$) and the rate factor ($\beta$). The rate factor can be calibrated to specific scenarios to provide more accurate results. The mass passing the aperture would be added to the zone below or added to the total mass passing all decks if there are no decks below the current one.

$$k_j = 80(e^{(-\beta d_{50}/Ap)} - e^{(-\beta)}d_{50}/Ap) \tag{5}$$

Material velocity can be determined by the inclination, frequency, and the throw of the screen. The input to the model was the angle ($\alpha$) for every section. In the model, the incline of the screen changes the transport velocity of the rock material. Since a higher velocity would make the layers thinner, it would also change the stratification and passage through the screen decks. The velocity of the material was estimated with Equation (6). For a more detailed model description, see Soldinger [3] and Asbjörnsson et al. [2]. Figure 7 shows the mass balance between the different layers of vibratory.

$$v = (0.064\alpha + 0.2)(380R - 0.18)(0.095f\alpha^{-0.5} + 0.018\alpha - 0.38) \tag{6}$$

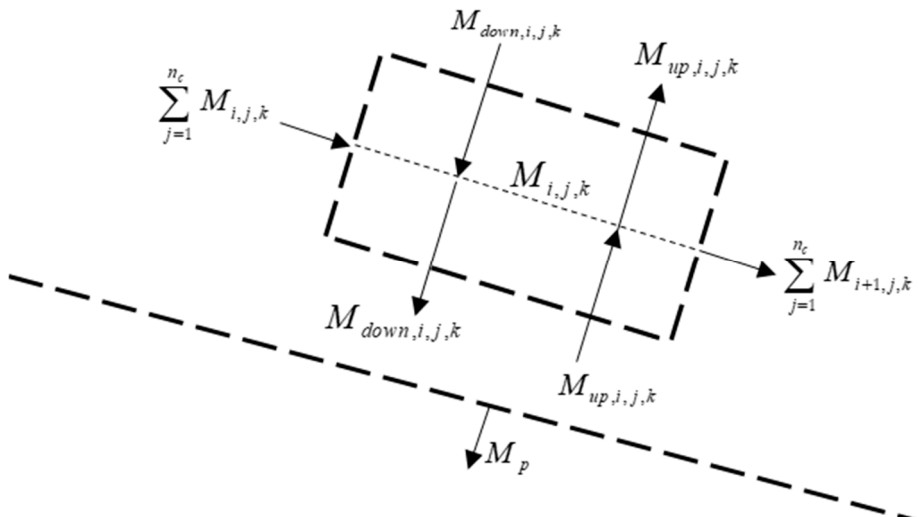

**Figure 7.** The mass balance between different layers of vibratory.

For the implementation of the model, a geometric design of the screen was configured, the feeding position was adjusted, the feed rate was fixed at 5 kg/s, and the size distribution was defined to match the DEM model.

## 3. Results/Discussion

The DEM simulation results for the vibratory screening process are shown in Figure 8, for which the feed rate increased from 4 kg/s to 6 kg/s. As the feed rate increased, a thicker, and approximately constant thickness material layer built upon the screen deck.

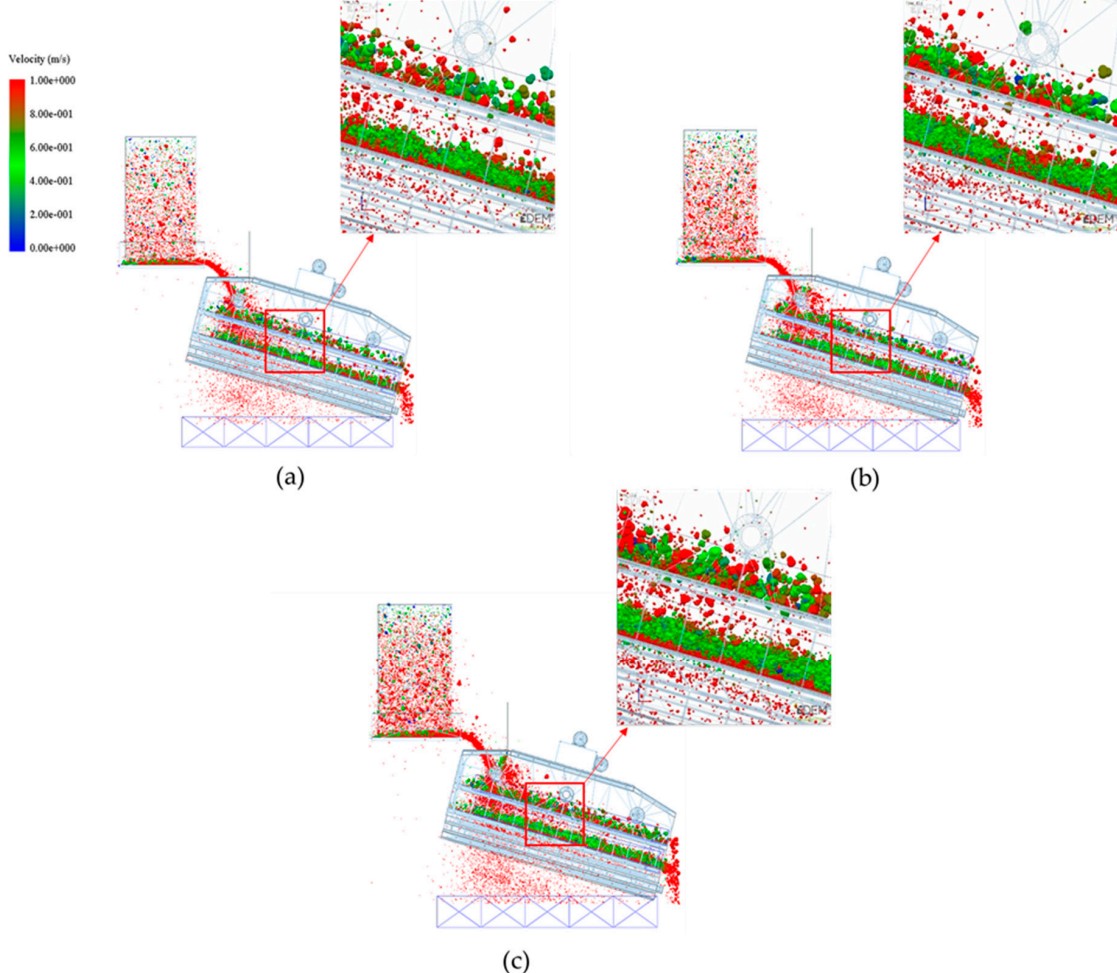

**Figure 8.** Discrete element method (DEM) simulation results for the vibratory screening process with three different feed rates: (**a**) 4 kg/s, (**b**) 5 kg/s, and (**c**) 6 kg/s.

The stratification process is different for various bed thicknesses. The stratification process that occurs during the screening process is expected to proceed in the vertical direction; thus, a thicker material bed requires more time for stratification. One way to adjust the stratification time is to control the particle velocity during the screening process by adjusting the inclination and frequency of the vibratory screen.

Stratification has the largest impact on screening efficiency. Stratification can be affected by different factors, such as material segregation, which itself depends on the physical particle characteristics, such as the particle size, shape, density, and surface texture. Particle size has a significant effect on segregation and stratification. Smaller particles can move towards the screen through the gaps between large particles, which is a key factor in screening efficiency.

In this paper, different sections of the vibratory screen were analyzed by using bin numbers to separate the sections. Material densities are simulated using three different feed rates. Figure 9 shows the percent of the particles (partition number) that passed through the different sections. The low-density material had a better passage than the high-density material.

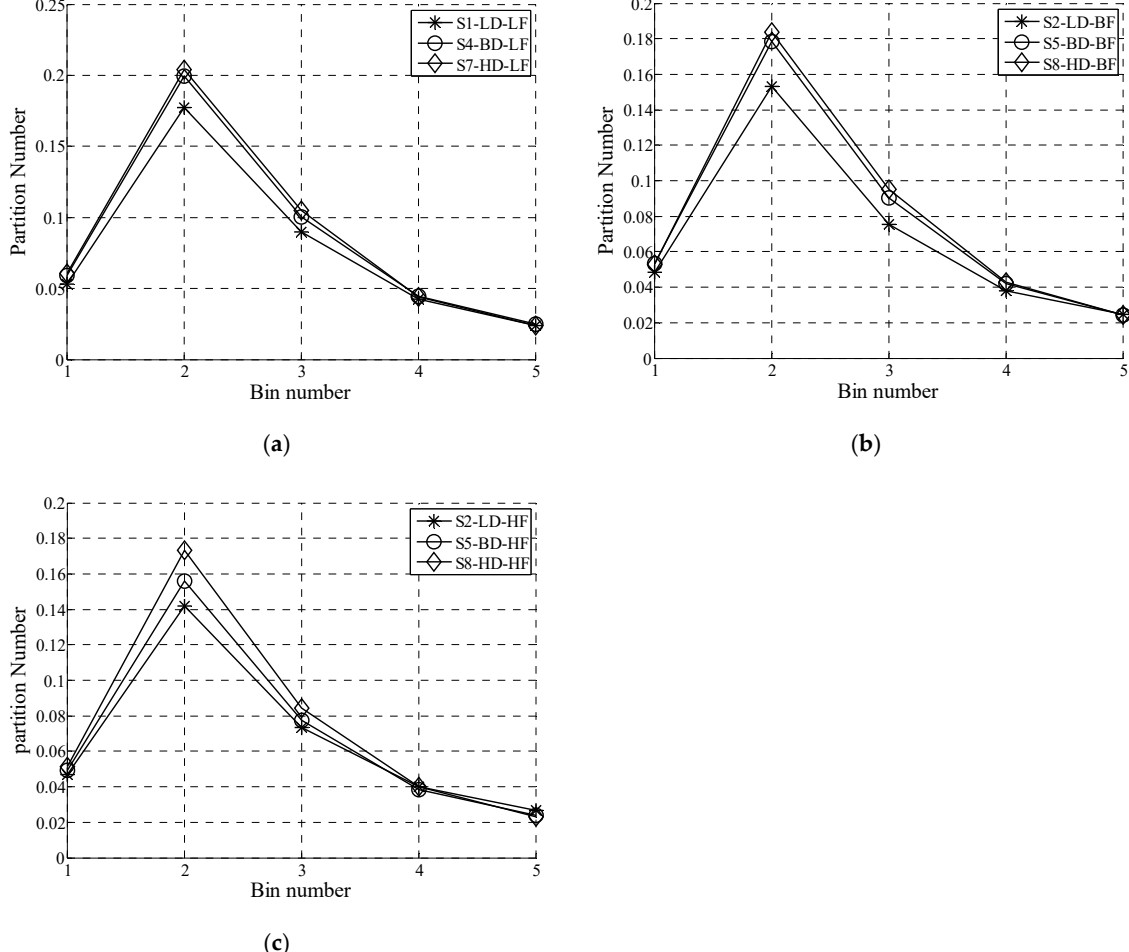

**Figure 9.** Passage rate in different screen sections for different densities and feed rates (LD: Low-density material, BD: Between density material, and HD: High-density material. (**a**) Low feed rate, 4 kg/s, (**b**) between feed rate, 5 kg/s, and (**c**) high feed rate, 6 kg/s.

For further investigation, the material discharge was studied to compare the numbers of undersized material in the overflow for both decks. The average particle diameter in the overflow material can be seen in Figure 10. The average diameter was lower for the low-density material for both decks, which means that the simulation of the low-density material had more undersized material in the overflow than high-density material. However, this difference was smaller on the upper deck than under the deck, since the upper deck had a larger aperture, and there were more chances of smaller particles passing through the screen deck.

Upon increasing the feed rate, the average particle diameters decreased for both decks (Figure 10), as increasing the number of particles in the simulation resulted in a thicker bed (Figure 8) and decreased the probability that smaller particles could contact the screen deck.

The particle velocity during the screening process has a huge impact on screen efficiency. The more particles that remain on the screen surface, the greater is the probability that these particles will pass through the screen deck. The average particle velocity for both screen decks is shown in Figure 11. The variation in the velocity for different material densities was not significant, and the higher-density particles had a slightly higher velocity. As Figure 11 shows, increasing the feed rate decreased the particle velocity along the screen deck as the number of collisions between the particles increased due to more material being on the screen deck.

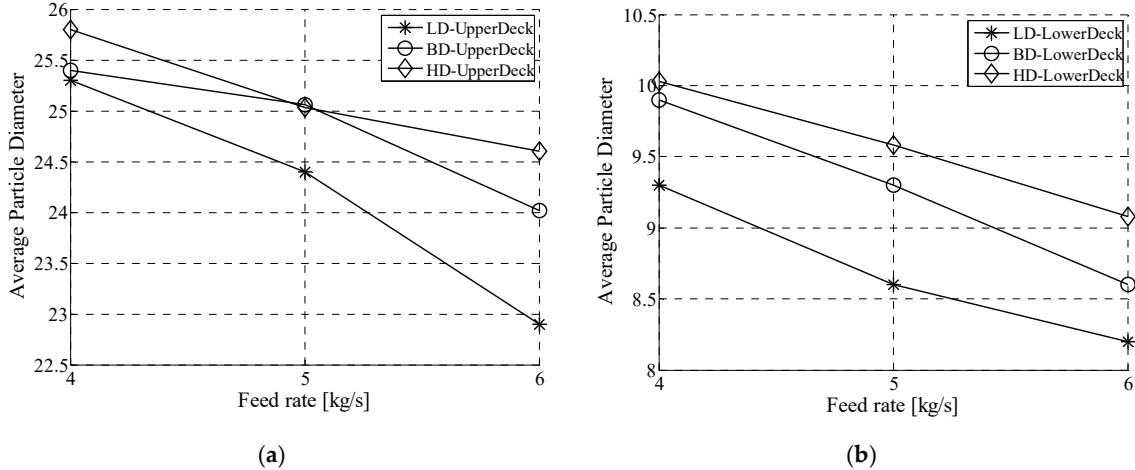

**Figure 10.** Average particle diameter in overflow material. (**a**) Upper deck. (**b**) Lower deck. (LD: Low-density material, BD: Between density material, and HD: High-density material)

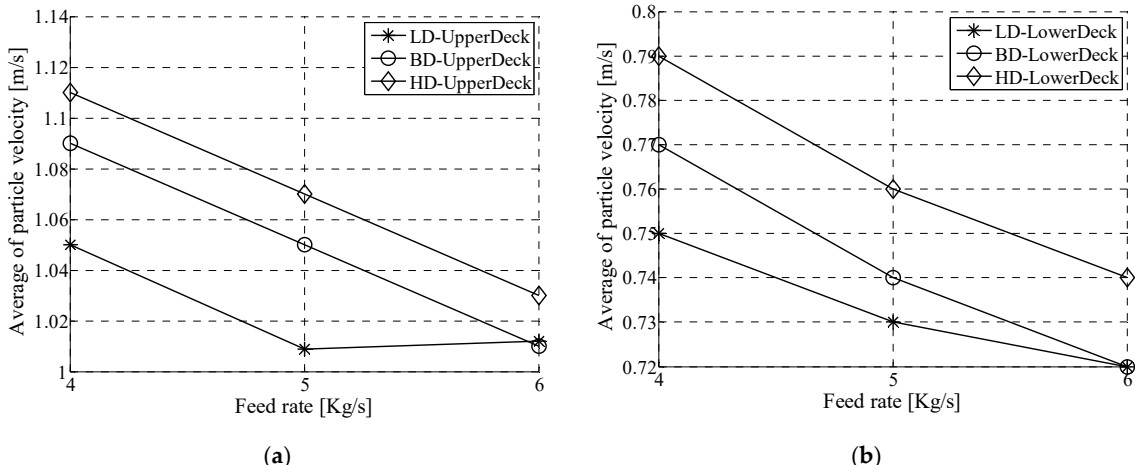

**Figure 11.** Average particle velocity for different material densities, (**a**) upper deck, (**b**) lower deck. (LD: Low-density material, BD: Between density material, and HD: High-density material)

Stratification can be affected by the difference in material densities; thus, a simulation with two different material densities was performed to determine the size of the effect of the material density on the screening performance. To generate the same volume of material for different densities, numbers of particles were used in the simulation instead of generating the materials by mass. As a result, the particle factory generated different numbers of particles to achieve a specified mass.

Figure 12 shows the total number of particles that passed through the screen deck in different sections. More high-density particles passed through the screen deck than for the low-density material. As Figure 12 shows, at the discharge point of the upper deck, there was almost the same number of particles for both the high- and low-density materials. However, the total number of low-density particles was greater at the discharge point in the lower deck.

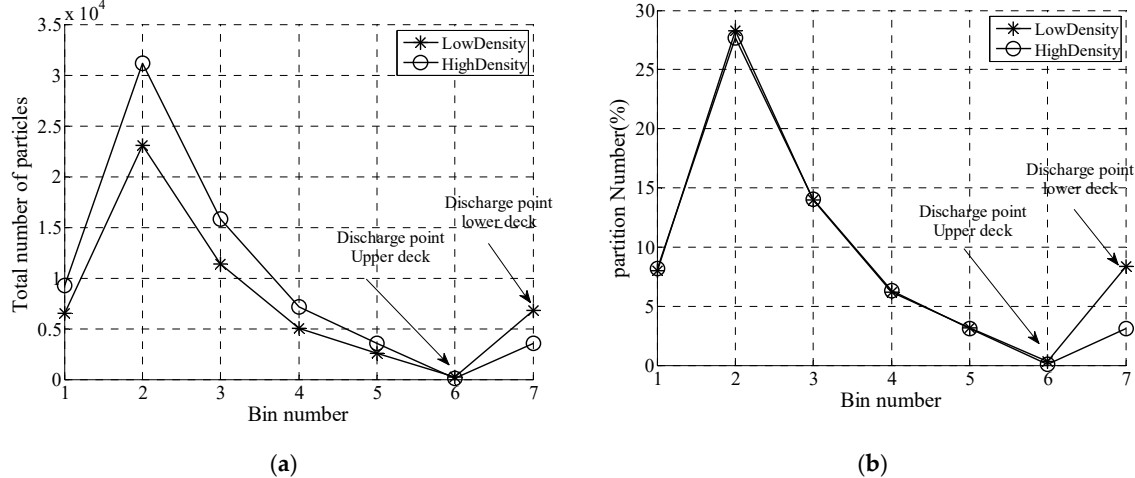

**Figure 12.** Total number of particles passed through screen deck in different sections with the feed for mixed particle densities. (**a**) Total number of particles. (**b**) Partition number.

*Model Comparison*

The mechanistic screen model presented by Soldinger [20] was used to study the theoretical passage rate along the screen deck. The factor β was calibrated to the value of 8.1, compared to 6.7 in Soldinger's original work [17]. A comparison was made by changing the material density of the incoming feed and estimating the passage rate at different positions in the same manner as used for the DEM model. The feed rate was fixed at 5 kg/s, and the size distribution was the same as the DEM simulation, to make the simulations comparable.

In Figure 13a, the result from model simulation and data from DEM are shown; Product 1 is the course product, Product 2 is the middle product, and Product 3 is the fine products at different intervals. As Figure 13a shows, the particle size distribution for the middle and fine products was around the same range. However, in the DEM simulation results, there was a significant portion of the middle size fraction, which remained in the coarse product fraction. This could be due to a few reasons; first, the shape of the particles defined in both simulations was different, since the particles in the DEM simulation had a more spherical shape, and the second, it could be due to the effect of friction. Friction between particles and between particles and the screen deck is different based on different rock materials used and screen deck material. In the case of the DEM model, this was covered by the Hertz–Mindlin contact model; in the mechanistic model, it was not explicitly formulated as an input variable.

The effect of different densities in the passage rate was studied, which can be seen in Figure 13b. The mechanistic model and the DEM simulation showed similar trends. The simulation result showed that the material with higher density had more of a chance to passage during the screening process into the first two bins.

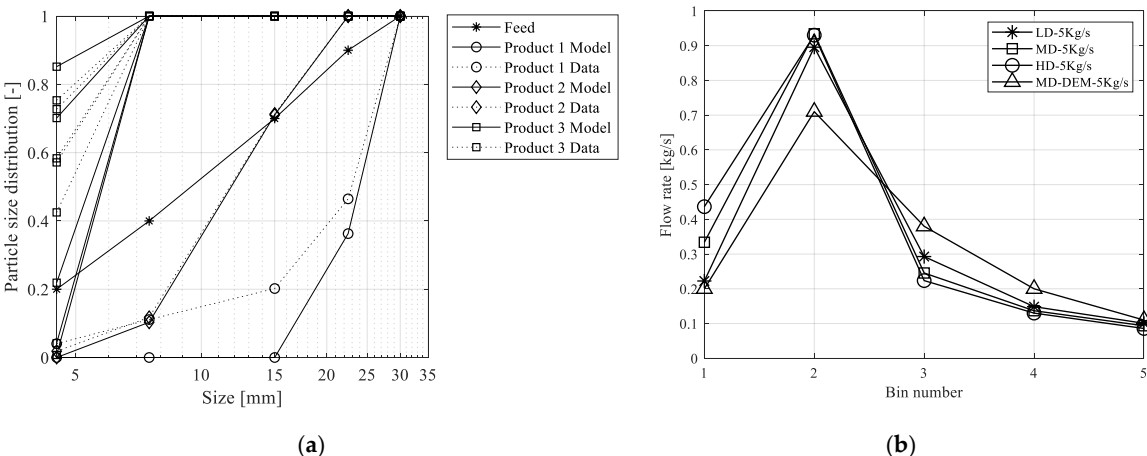

(**a**)    (**b**)

**Figure 13.** (**a**) Passage result from model simulation (product model), and DEM simulation (product data). (**b**) Passage rate in different screen sections by using screen model simulation.

## 4. Conclusions

Several main conclusions can be drawn from this study.

1.  DEM is a powerful tool for calculating the overall efficiency and the product size distribution, while also enabling the analysis of important parameters, such as the size fraction (by sampling from any part of the screen); particle tracking; and the observation of bed material, which helps in stratification analysis. Therefore, DEM can provide a better understanding of different process parameters, such as how various particle densities have an effect on the stratification process and screen efficiency.
2.  The increase in the passing percentage of small undersized particles mainly occurred at the upper deck, and as a result, the particle bed was thicker at the lower deck, which means that the stratification process mainly affects the screen efficiency in the lower deck.
3.  In the simulations, the passage rate for the low-density material was lower than for the high-density material, since the low-density material had a lower stratification rate compared to the high-density material.
4.  In the stratification of materials with various densities, it is easier for the high-density material to move vertically through the particle bed. Thus, the high-density material has a higher probability of passage.

## 5. Future work

In future work, experimental tests will be performed using a laboratory-scale vibrating screen that was used in all of the simulations in this paper.

Several parameters can be studied by using DEM, and experimental tests can be performed, such as determining the effect of the particle size distribution on the screen efficiency. The particle shape is an interesting factor that should be investigated to determine the significance of its effect on the stratification process.

**Author Contributions:** Conceptualization, A.D. and G.A.; methodology, A.D. and G.A.; software, A.D.; validation, A.D.; formal analysis, A.D.; investigation, A.D.; resources, A.D. and G.A; data curation A.D.; writing—original draft preparation, A.D.; writing—review and editing, G.A., E.H. and M.E.; visualization, A.D., G.A., E.H. and M.E.; supervision, G.A., E.H. and M.E.

**Funding:** The research presented in this paper was funded by Chalmers Area of Advance Production.

**Acknowledgments:** Thanks to all my colleagues at Chalmers Rock Processing research group for your support.

**Conflicts of Interest:** The authors declare no conflict of interest.

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
