# Peer review of "Application of the Discrete Element Method to Study the Effects of Stream Characteristics on Screening Performance"

_minerals, doi:10.3390/min9120788_

Round 1
Reviewer 1 Report
The manuscript concerns a key-issue of the screening process and is very important from the scope of screening efficiency assessment. Article is well written and it is easy to follow the main scope, however some concerns appear during the reading.
Why the double-decked screen was used in DEM simulations? It is not clear esspecially when an average feed size is 6.25 mm, and apertures of individual decks are 25x25 and 10x10. In the "results" section the Authors have not analysed the obtained results from the scope of performance of each deck searately.
Additionally some more detailed characteristics of the feed particle size could be presented (i.e. PSD curve or st. dev).
There are also no explanation (Fig. 11(a)) how the reader should understand the categories "Prod 1 data", "prod 2 data", "Prod 3 data" - how the "data" points were obtained?
In section 2.3. "Modeling" some descriptions of formulas should be provided. In example there are no descriptions for symbols used in formula (4) and Fig (5). Line 140: what does the "rate_beta" means? Figure 7 - please provide descriptions for all legends.
It is not clear if the formula (5) is a proposal of Authors or a citation?
Section "Conclusions": first one is a general statement, quite obvious. Please reformulate it to make it more specific to the scope of paper
There are also some minor issues that should be improved:
line 28: operational instead of operation ? lines 30 and 38: finer instead of smaller line 43: phrase "Granular material with different material properties..." sounds a bit weird, suggest to remowe second "material" line 75: careen ? line 118: Table 1: please provide some information on how the user can define the particle diameter. line 139: Eq(5) instead of Eq(6) lines 164 and 236: kg/s instead of Kg/s line 240: coarse product line 266: lower instead lessSpelling errors:
line 139: exchanged line 251: two line 267: rate, comparedAuthor Response
Please see the attachment.

Reviewer 2 Report
Major Comments on the Article
The article attempts to study effects of particle density and flow rate on stratification and screening performance in a vibratory screen. Several aspects in the model setup, results and discussion are missing in the article. Below are some comments:
Line 94-95: The authors state “The test plan is designed to determine how different material densities affect the screening efficiency by altering the feed rate.” It is unclear what this means. How is effect of material density studied by changing the feed rate? Line 103: The authors state “Very different results are obtained using one spherical particle in the simulations versus multi-sphere particles.” Is this verified as a part of this work? Line 101: “Particles with user-defined diameters are employed to match the planned simulations.” What are the diameters/particle size distributions used? This is a key variable that influences interpretation of the results. Line 106-107: The statement “The material is a rock with three different solid densities of 2900 kg/m3, 2500 kg/m3 and 2100 kg/m3.” is unclear. Does this mean each of the spheres in Figure 2 has different density? The statement “The separation of these particles may represent the essential sorting operation that occurs in a typical screening process.” is unclear as well. Does this mean the simulation has particles similar to the rock structure (figure 2) but in various sizes? This needs to be clearly explained. Line 116: The authors state “simulations reach steady-state after 6 s.” The statement needs to be substantiated with associated plots that show evolution of number of particles with time. Equation 4: This equation is not well explained. What do the subscripts j, nl, BP stand for? Equation 5, 6: Can the model proposed by Soldinger be used directly in this work? Model parameter used remain unchanged. This discussion is important. Line 168-169: The authors state “As the feed rate increases, an improvement in the particle screening performance results in better material transportation and a thicker and approximately constant-thickness material layer, as shown in Figure 6.” The conclusion drawn by the authors is not evident from the figure. Figure 7: Figure 7 shows ‘partition number’ on y-axis, which is not explained in the article. In addition, legends are not explained. It is unclear what and how results are drawn from this. Figure 8: Results from Figure 8 cannot be interpreted as size distribution of the feed is not given in the paper. It is unknown what is the particle bed made of in these simulations? Figure 9: It is unclear if these are obtained from DEM simulations or empirical equation What do ‘bin numbers’ in these figures refer to? Are they particle size bins or sections in the screen? Line 213-214: The authors state “increasing the feed rate decreases the particle velocity, because the number of collisions between the particles increases as there is more material on the screen deck.” One might argue that increased collisions should increase the velocity. Line 221-223: The author state “To generate the same volume of material for different densities, numbers of 221 particles are used in the simulation instead of generating the materials by mass. As a result, the particle factory generates different numbers of particles to achieve a specified mass.” To understand effect of material density one would want to have same number of particles with different densities. Simulation settings can typically be changed to achieve this easily.
Minor comments
Line 30: What frequency and amplitude? Specify if these are the frequency and amplitude of vibrations. Line 31: ‘aperture’ is not an equipment parameter. ‘aperture size/shape’ is the relevant parameter Line 31/32: operational strategy and operational factors typically mean the same. Suggest distinguishing equipment parameters and stream characteristics more clearly. Line 45: The authors state “The two most important material properties that can affect 44 segregation are the particle shape and density.” What about size? Line 51-53: Rewrite the sentence “Clément, Rajchenbach [9] performed experiments on the motion of 51 single particles of different sizes in a bed of equal sized particles in a rotating cylinder by using an image-processing device.” Figure 1: Add reference Line 97: Specifically mention different feed rates used Table 1: Improve Table 1, make it more readable Table 1: Add references for DEM simulation parameter values used Figure 3: Improve resolution Equation numbers are mismatchedAuthor Response
Please see the attachment

Reviewer 3 Report
General comments:
The research paper is well written and the experiments performed are commensurate to the scientific standards.
On the paper:
The objective of the study was to characterize the influence of screen performance. It looks at the effects of material densities on stratification and passage. The subject area is important not only in the minerals industry, but all sorts of industries such as agriculture and pharmaceuticals among others. The use of Discrete Element Method (DEM) presents the novelty of the paper.
Concerns:
The author has however relied heavily on a limited amount of references (not quantitatively, but qualitatively). Particularly, the implication that because only one author (Soldinger) did not take into consideration the effect of density on stratification is wanting, since this was 20 years ago. This issue is no longer relevant. Besides, today's scientific studies of dry granular material no longer rely solely on collision models such as DEM, but includes continuum methodologies for comparison. This is because it is difficult to draw a line between when or where physico-chemical properties of materials transform from discrete to continuum streams.
Overall assessment:
The paper can be improved by slightly widening the reviewed models by reviewing more recent publications since there was only one paper written in 2019 and only a few from 2015 onward. I also recommend that the author also explores the effects of the same parameters (particle sizes, densities, and flow velocities) and compare the results to at least give a general view on what happens during transitions from larger solids to much smaller fragments.
Decision:
I therefore recommend that the paper be worked on more thoroughly before accepting to minerals.
Round 2
Reviewer 3 Report
The article has met a bare minimum. I therefore recommend its publication in the journal.